# Spatial Epidemiology and Its Role in Prevention and Control of Swine Viral Disease

**DOI:** 10.3390/ani14192814

**Published:** 2024-09-29

**Authors:** Juan Qiu, Xiaodong Li, Huaiping Zhu, Fei Xiao

**Affiliations:** 1Key Laboratory of Monitoring and Estimate for Environment and Disaster of Hubei Province, Innovation Academy for Precision Measurement Science and Technology, Chinese Academy of Sciences, Wuhan 430077, China; lixiaodong@whigg.ac.cn (X.L.); xiaof@apm.ac.cn (F.X.); 2Laboratory of Mathematical Parallel Systems (LAMPS), Department of Mathematics and Statistics, Centre for Diseases Modeling (CDM), York University, Toronto, ON M3J1P3, Canada; huaiping@yorku.ca

**Keywords:** spatial epidemiology, geographical distribution, risk identification, mapping, swine viral disease

## Abstract

**Simple Summary:**

Spatial epidemiology, integrating traditional epidemiology, geography, statistics, environmental science, and ecology, provides a comprehensive framework for analyzing the spatial dimensions of health and disease. This interdisciplinary approach enhances the development of effective public health strategies and interventions. However, its multifaceted nature can bring complexities in practical application. Using the case of spatial epidemiology in swine viral diseases (SVDs), we illustrate the objectives, methodologies, and essential considerations for the application of spatial epidemiology, which we hope to offer as a comprehensive reference for researchers in this field.

**Abstract:**

Spatial epidemiology offers a comprehensive framework for analyzing the spatial distribution and transmission of diseases, leveraging advanced technical tools and software, including Geographic Information Systems (GISs), remote sensing technology, statistical and mathematical software, and spatial analysis tools. Despite its increasing application to swine viral diseases (SVDs), certain challenges arise from its interdisciplinary nature. To support novices, frontline veterinarians, and public health policymakers in navigating its complexities, we provide a comprehensive overview of the common applications of spatial epidemiology in SVD. These applications are classified into four categories based on their objectives: visualizing and elucidating spatiotemporal distribution patterns, identifying risk factors, risk mapping, and tracing the spatiotemporal evolution of pathogens. We further elucidate the technical methods, software, and considerations necessary to accomplish these objectives. Additionally, we address critical issues such as the ecological fallacy and hypothesis generation in geographic correlation analysis. Finally, we explore the future prospects of spatial epidemiology in SVD within the One Health framework, offering a valuable reference for researchers engaged in the spatial analysis of SVD and other epidemics.

## 1. Introduction

Spatial epidemiology encompasses the description and analysis of geographic variations in disease concerning demographic, environmental, behavioral, socioeconomic, genetic, and infectious risk factors [1]. It integrates place and location into study designs and models, employing spatial approaches to investigate the distribution and determinants of epidemics. Consequently, it typically relies on Geographic Information Systems (GISs) and spatial statistics. GIS maps offer visualization tools for exploring the spatiotemporal distribution of disease outcomes, associated social factors, and environmental exposures. Analytical methods, such as estimating disease risks and identifying spatial disease clusters, provide a detailed statistical perspective on the spatial distribution of diseases. Spatial and spatiotemporal models aid in understanding, measuring, and analyzing disease syndemics, as well as the social, biological, and structural factors associated with them across space and time [2].

Initially, spatial epidemiology was primarily applied to vector-borne diseases [3], as their spatial distribution is often constrained by the geographical range of vectors or reservoir hosts, as well as their habitat preferences. Given that the geographic distribution of vectors or hosts is often expressed in ecological landscapes, it is also referred to as landscape epidemiology [3]. Additionally, it intersects with the research topics and methodologies of medical (or health) geography [4]. Here, we contend that spatial epidemiology focuses on understanding how the distribution of disease or health outcomes varies spatially and the factors contributing to this variation. It entails analyzing geographical patterns of disease incidence, prevalence, or mortality to uncover underlying causes, risk factors, and potential interventions.

The role of spatial epidemiology in disease prevention and control has been summarized as visualization (disease mapping), detection of clusters/hotspots, and identification of risk factors (geographical correlation studies) [1,5]. It has been widely applied to vector-borne diseases, zoonoses, and emerging infectious diseases [5,6,7,8,9,10,11]. In this article, we provide a comprehensive review of the application of spatial epidemiology to swine viral diseases, highlighting commonly used spatial analysis methods and their roles in disease control. These applications are categorized into four primary objectives: visualizing and elucidating spatiotemporal distribution patterns, identifying risk factors, risk mapping, and tracing the spatiotemporal evolution of pathogens. We elucidate the technical methods, software, and considerations necessary to accomplish these objectives. Additionally, we discuss key issues such as the theoretical basis of geographical correlation analysis, alternative interpretations of ecological fallacies, and conclude by offering perspectives on the future direction of spatial epidemiology in this field.

## 2. Visualizing and Elucidating Spatiotemporal Distribution Patterns

### 2.1. Disease Mapping

Here, disease mapping specifically refers to using observed data to visually display historical or current health or disease conditions, including mapping point locations of cases, incidence rates by area, standardized rates, and other related information [12]. This form of representing health or disease distribution is more intuitive compared to numbers, tables, and text, and it helps to identify subtle distribution patterns. For example, Vinod Kumar Singh et al. [13] mapped the temporal and spatial distribution of the CSF virus genotype circulating to present the current status of CSF in India.

It is important to note that when original disease data involve small areas or limited samples, directly mapping disease incidents based on raw crude rates or relative risks often lacks stability in depicting the “geographical distribution of diseases” [14]. This approach may present misleading images of the actual relative risk [15]. Hence, Bayesian disease mapping [16] emerged, offering the advantage of generating smoothed disease rates or risks, enabling the creation of maps that better reflect the true relative risk. Among these methods, the Bayesian hierarchical (BH) model is the most widely used. Thibault Saubusse et al. [17] mapped the predicted CSF seroprevalence in northeastern France using a spatial BH model. Clément Calenge et al. [18] mapped the posterior mean probability of first consumption of the vaccine by non-immunized wild boar against CSF using the BH model, accounting for population dynamics. Figure 1 is an example of a mapping study of the prevalences for wildlife PRV infections in the Federal State of Brandenburg [19]. Figure 1a shows the observed prevalences of the PRV infections of wild boars per municipality or their exclaves based on aserological survey of the year 1993. Figure 1b shows the smoothed prevalences for PRV infections, with smoothing to the overall median using the BH model.

### 2.2. Spatiotemporal Pattern Recognition

After mapping the disease, initial observations of spatiotemporal characteristics may be possible with limited data; however, these observations are often based on empirical experience and lack statistical validation. With larger datasets, it becomes crucial to employ spatial analysis methods to identify and elucidate spatiotemporal patterns. This process, known as spatiotemporal pattern recognition or exploratory spatial data analysis [2], involves using spatial analysis techniques to discern patterns over time and space. These tools may involve both visual representations and statistical inferences to analyze data distributions. They are employed to detect spatial patterns, identify outliers and clusters, formulate hypotheses for subsequent statistical tests, and guide the development of spatial regression models. Similarly to exploratory data analysis, where understanding the underlying data distribution is vital for selecting appropriate statistical tests and formulating models, exploratory spatial data analysis is fundamental to spatial modeling and hypothesis testing [2].

Based on the spatial data types—namely, point data, line data, and area data (polygons)—we have compiled a summary of commonly employed spatial statistical methods, including their advantages, applications, associated software, and relevant case studies from the literature (see Table 1).

In spatial epidemiology, the Mean Center tool is utilized to identify the geographic center, or center of concentration, of a set of outbreak points. This analysis is often a preliminary step for tracking the movement of the center over time. The design of time segments is critical when employing the Mean Center tool, as it directly influences the interpretation of the results. Improperly defined time segments—whether too granular, too broad, or neglecting key periods—can obscure the spatiotemporal transmission characteristics of the epidemic. In cases where the number of center points is relatively small and there is a discernible spatial movement trend, the temporal shift of the outbreak center can often be visually assessed. For example, using the Mean Center tool within the ArcView Geographic Information System software, researchers determined the shift in the epidemic center of ASF in the Russian Federation from 2007 to 2012. This analysis revealed a trend of diffuse ASF transmission in regions adjacent to the primary and secondary epidemic zones [20].

Furthermore, directional analysis tools may be necessary to assess systematic patterns in outbreak spread. The direction test [44] helps determine whether outbreaks exhibit a systematic directional spread. In cases where the direction of spread is not immediately apparent, the outbreak sequence is often divided into time segments (e.g., weeks or months). Initially, a Mean Center analysis identifies outbreak centers for each time segment. Subsequently, directional analysis evaluates whether these centers display statistically significant directional patterns. For instance, Shao Qihui et al. [29] employed the direction test to investigate whether the monthly outbreak centers of ASF in Vietnam, identified using the Mean Center tool, exhibited directional spread.

The colocation quotient statistic is utilized to investigate potential associations between ASF outbreaks in domestic pigs and wild boars, such as determining whether an outbreak in domestic pigs is linked to infections in nearby wild boars [25]. Beyond the standard epidemiological spatial analysis tools available in existing software, there are also specialized spatial analysis methods developed for specific analytical purposes. For instance, M. Lange et al. [21] devised an algorithm to differentiate between endemic and non-endemic patterns of case occurrence, which was employed to assess whether ASF had become endemic in the wild boar population in southern Russia, based on space–time correlation hypotheses.

When dealing with data involving line-type observations, such as livestock transportation routes, network analysis becomes pertinent. Alfredo Acosta et al. [36] constructed a directed network using pig premises—including farms, traders, industrial zones, and markets—as nodes. They analyzed this network by computing various metrics such as betweenness, closeness, clustering coefficient, degree, density, diameter, average shortest path, and the giant weakly and strongly connected components. This analysis aimed to uncover spatial characteristics of the swine network, identify trading communities within it, and assess the network’s role in the spread of CSF using the k-statistic.

Statistical analyses of outbreaks are frequently performed by administrative units (e.g., community, town, city, state) to identify hotspot areas, with spatiotemporal scan statistics and clustering being commonly employed methods. For example, SaTScan software was used to detect spatial–temporal clusters of high PRRSV seroprevalence in China from 2017 to 2021 [39]. Similarly, Daniella N. Schettino et al. [45] utilized SaTScan to identify clusters indicative of the predicted risk for ASF introduction into Kazakhstan.

In many instances, a combination of spatial analysis methods is employed to elucidate the spatiotemporal characteristics of a disease. For example, the maximum spatial association radius of ASF cases can be determined using Ripley’s K function. This radius is then used as the maximum extension for the spatial window in space–time scan statistics, which assists in identifying and delineating spatiotemporal clusters of ASF in wild boars in the Russian Federation from 2007 to 2013. Subsequently, the mean reproductive ratio (R0) for each spatiotemporal cluster was estimated [22]. To delineate the spatiotemporal distribution pattern of ASF in Sardinia, I. Iglesias et al. [23] employed a range of spatial analysis methods, including spatial autocorrelation and spatiotemporal clustering. Figure 2 is an example of a combination using directional analysis and spatiotemporal scan statistics to uncover the dynamics of spatial diffusion and spatiotemporal aggregation characteristics of ASF in Vietnam by [29]. Figure 2A shows the monthly outbreak centers and average spread direction of the ASF in Vietnam, February 2019–March 2022. Figure 2B shows the spatiotemporal high-risk clusters of the ASF in three phases.

## 3. Risk Factors Identification

Identifying risk factors involves elucidating the variables that influence the spatial or spatiotemporal structure, distribution, and transmission patterns of health-related conditions. This process is often achieved through geographic correlation studies [1]. We have summarized and listed the geographic analysis methods commonly used in SVD, as outlined in Table 2.

Environmental factors associated with SVD are extensively analyzed using statistical methods. Among traditional methods, logistic regression [46,47,61] is widely employed. For instance, multivariable logistic regression analysis was used to analyze the risk factors such as farm size, geographic location, before or after the ASF outbreaks and PRV purification associated with the PRRSV serological status of pig farms [39]. Cross-correlation analysis has examined the relationship between climate variables (rainfall, wind speed, temperature, vapor pressure, and relative humidity) and CSF outbreaks in China from 2005 to 2018, revealing that low relative humidity and high wind speed are significant meteorological factors associated with CSF outbreaks [38]. However, epidemiological data for SVD often exhibit non-independence in both spatial and temporal dimensions, rendering conventional statistical methods like ordinary linear regression unsuitable for hierarchical data. Instead, multilevel statistical models, such as Hierarchical Linear Models, Random Coefficient Models, Variance Component Models, and Mixed-Effects Models, are necessary [62]. Some indicative approaches, such as the Generalized Linear Logistic Regression [25], the generalized mixed-effects model [48,51], Bayesian hierarchical models [49,54,63], Bayesian spatial mixed multivariable logistic regression [52], and the mixed-effects Poisson regression model [50,53], have been widely used in SVD. For example, Generalized Linear Logistic Regression was used to explored socioeconomic and environmental factors influencing ASF outbreaks in the Russian Federation from 2007 to 2020 [25]. Additionally, in this study, Moran’s I was employed to assess spatial autocorrelation in both the response variable and residuals, ensuring that the model accounted for spatial dependencies.

The spatial factors incorporated into SVD statistical models (Table 2) may vary depending on the spatial scale, the understanding of the disease epidemiology, the quantity and quality of available data, and the background and experience of the modelers. The identified risk factors can be categorized into two main types: natural geographic factors and human geographic factors. Specifically, natural geographic risk factors include aspects of climate, such as humidity and wind speed, as well as temperature [38,54]. They also encompass landscape and habitat features, including agriculture, forestry, and open-canopy pine, prairie, and scrub habitats [46,48,64]. Additionally, wildlife risk factors such as wild boar density and the appearance of raccoons, raccoon dogs, and crows are identified [48,49,51]. Human geographic risk factors can be divided into several categories. Farm and animal density factors include pig density and pig farm density [39,50]. Transport and network factors cover truck networks and animal movement networks [47,54]. Relationships with infected areas involve proximity to previous SVD incidents, the importation of live pigs from SVD-affected regions, the volume of pork products imported from affected regions, and the presence of a common border with an SVD-affected region [25,48]. Lastly, the individual level factors include age and personal consumption [52,53].

In spatial epidemiology, mathematical models incorporate spatial attributes such as individual locations and contact structures between individuals [65]. This allows for the study of outcome distributions under various scenarios, including the promotion of vaccination, improvement of health systems, modification of social structure and behavior, and implementation of health interventions. For example, the spatially explicit, individual-based (SEIB) model, which integrates information on population distribution, movement, and environmental structure, has been extensively applied in various scenarios of SVD [55,66,67,68,69]. An SEIB model for ASFV was developed to estimate the impact of carcass-based transmission on the persistence of ASFV in European wild boar [55]. This study utilized a landscape divided into 5 km × 5 km grid cells, covering a total area of 6000 square kilometers. The model accounted for variations in the social structure of wild boar populations within each grid cell, as well as their dispersal between cells. Martin Lange et al. [67] used the SEIB model to evaluate the effectiveness of various spatiotemporal vaccination strategies for CSF. The model accounts for both regional scale dynamics and individual infection processes. Other typical mathematical models applied in SVD include Be-FAST [56], the North American Animal Disease Spread Model (NAADSM) [57], the spatial simulation in a gridded landscape based on cellular automata and susceptible–exposed–infected–recovered (SEIR) model [58], and the Monte Carlo simulation method. These models focus on simulating the transmission dynamics of SVD (Table 2). For instance, the Monte Carlo simulation method has been employed to model the daily transmission of FMDV [70,71,72], CSF [31,73,74,75,76,77,78,79,80,81,82,83,84,85] and PRRSV [86] between farms (or wild pig herds), as well as the effects of control measures, including vaccination [71,87]. The rate of infection for each livestock farm is typically modeled using a spatial transmission kernel [87,88,89,90,91], which illustrates how the infection risk between susceptible and infected farms varies with distance. This kernel illustrates the influence of spatial heterogeneity on the rate of epidemic spread. Since all transmission and control mechanisms are spatially defined based on farm locations, this is termed a spatial, dynamic, and stochastic epidemiological model, known as Interspersal models [59,92].

Recently, artificial intelligence (AI) has been applied to predict SVD. The Extreme Gradient Boosting (XGBoost) machine learning model, combined with Synthetic Minority Over-sampling Technique, was used to generate weekly PEDV predictions for sow farms in the U.S., considering recent animal movements, current PEDV status of farms, environmental factors (such as average temperature and humidity), and land use characteristics (including hog density and land use proportions) [60]. Boosted regression trees (BRT) [42] were utilized to relate the presence or absence of PRRS outbreaks at the sub-district level to demographic characteristics of pig farming and other spatial epidemiological variables, such as human population density and the number of farms with breeding sows.

## 4. Risk Mapping

Identifying risk factors is crucial for implementing targeted control measures and for assessing disease risks across a given area. Geographic correlation analysis, which accounts for both direct and indirect factors, enables comprehensive risk assessments and predictions throughout the area of interest based on revealed geographic risk factors [93,94,95]. Therefore, after statistical models are commonly used to identify risk factors, risk mapping is often carried out. For example, Weerapong Thanapongtharm et al. [42] utilized autologistic multiple regressions and boosted regression trees (BRT) to quantify the relationship between PRRS presence and various risk factors, subsequently mapping the predicted probability of PRRS occurrence in Thailand. Anastasia A. Glazunova et al. [25] mapped the predicted probability of having an ASF outbreak in domestic pigs within the Samara oblast (Russian Federation), based on socioeconomic and environmental factors revealed by Generalized Linear Logistic Regression. Additionally, some methods are used to directly perform risk mapping. For example, William A de Glanville et al. [96] predicted the continental distribution of suitability for ASF persistence in domestic pig populations in Africa, using spatial multi-criteria decision analysis, a method for determining the suitability of ASF based on risk factors obtained through a literature review. Additionally, the maximum entropy model has been extensively used to generate risk maps for SVD outbreaks, such as predicting suitable areas for ASF in wild boars across northern Eurasia [35] and mapping PRRS outbreak risks in swine farms in the Midwest region of the U.S. [97].

Similarly, after using mathematical methods to simulate the spatial transmission of SVD, risk mapping can also be further developed. For example, Beatriz Martínez-López and colleagues [31] utilized real pig farm distribution data in Bulgaria and employed Be-FAST to simulate the daily spread of CSFV within specific farms and the transmission of CSFV between farms. This model integrated a discrete-time stochastic Susceptible-Infectious (SI) framework, accounting for spatial location, the demographics of farms, and their contact patterns. They ultimately mapped the spatial distribution of CSFV in Bulgaria using the Kernel density function. Similarly, the dynamic transmission process of CSFV within and between pig farms was simulated by Be-FAST, with the results used to map the risk of CSFV introduction through spatial interpolation in the Segovia province in Spain [56]. At the prefectural level, either a distance-related gravity model or a neighborhood model for CSF has been developed for the entirety of Japan [94]. This model considered factors such as seasonality, agricultural damage data, county area, and forest area related to pig population size and density. It then calculated and mapped the infection risk for both pig and wild boar populations in each prefecture. This infection risk assessment was employed to determine the geographic scope of pig farm vaccination, aiding in the formulation of recommended vaccination policies.

When direct or indirect geographic spatial data are unavailable for the identified risk factors, spatial interpolation can be employed for risk mapping. For instance, Satoshi Ito et al. [51] utilized two spatial interpolation methods—inverse distance weighted (IDW) and Kriging—to create a probability map illustrating wild boar occurrence across Gifu Prefecture, Japan. While the study initially used generalized linear mixed models to identify factors associated with wild boar emergence (such as raccoons, raccoon dogs, and road density), accurately extrapolating these factors across the entire area using geographic correlation analysis is challenging probably due to the lack of spatial data on raccoon and raccoon dog distributions. It is crucial to note that spatial interpolation is unsuitable for non-continuous (discrete or unevenly distributed) health or disease elements, such as ASF outbreaks linked to farm locations and their culling or vaccination measures. Therefore, caution should be exercised when applying spatial interpolation in such contexts.

## 5. The Geographical Distribution of Pathogens and Their Spatiotemporal Genetic Evolutionary History

Spatial analysis extends into molecular epidemiology, integrating with genomics and genetics to examine the geographical distribution of pathogen lineages and their spatiotemporal genetic evolution. This approach, known as phylogenetic, phylodynamic, or phylogeographic analysis, presents the evolutionary patterns of pathogens in a more intuitive map form [98,99,100,101,102]. For swine viruses, current efforts primarily focus on mapping displays. However, as data volumes increase, the potential of spatial analysis in uncovering the spatiotemporal evolution of these pathogens merits further exploration.

## 6. Discussion

Epidemiological research at the geographic spatial scale can be significantly influenced by data availability and methodological approaches, such as information loss, sampling biases, cartographic confounding, and the modifiable areal unit problem [2]. These factors can lead to biased or questionable research results and interpretations. For instance, while maps are powerful tools for communicating spatial information, the method of disease mapping is contingent upon the objectives of the cartographers, which in turn affects the information accessible to readers. Variations in mapping techniques, such as differing spatial scales, resolutions, or color schemes, can produce different visual effects and potentially mislead interpretations [1,12]. Furthermore, the combined use of different spatial analysis methods is often employed to reveal spatiotemporal distribution or transmission patterns of diseases systematically. However, selecting multiple similar spatial analysis methods for spatiotemporal pattern recognition does not necessarily enhance the understanding of disease transmission patterns. Instead, it may result in inconsistent findings due to differences in statistical approaches or a lack of deep comprehension of the methodological assumptions. The expertise of cartographers or spatial analysts is crucial in these processes, posing a particular challenge for grassroots veterinarians.

Geographic correlation studies in spatial epidemiology are often regarded as hypothesis-generating [1], typically serving as an exploratory phase due to their limited support from established theories. These studies are usually conducted in the early stages to identify potential causes or patterns of disease, with the aim of guiding subsequent experimental research. However, it is important to recognize that geographic correlation analysis is not devoid of theoretical underpinnings. Insights can be drawn from fundamental geographic principles such as Tobler’s First Law of Geography [103,104], the Law of Spatial Heterogeneity [104,105,106], and Zhu’s Geographic Similarity [107].

Tobler’s First Law of Geography asserts that “everything is related to everything else, but near things are more related than distant things,” emphasizing the concept of spatial autocorrelation. This principle highlights that spatial phenomena exhibit stronger relationships when they are geographically closer, and it forms the foundation for spatial autocorrelation analysis and spatial interpolation, often represented mathematically through semivariance functions and distance decay functions. The Law of Spatial Heterogeneity, which states that “geographic variables exhibit uncontrolled variance,” underscores the variability and differences inherent in spatial phenomena—spatial heterogeneity merits further investigation. This heterogeneity is frequently modeled using techniques such as window Kriging and categorical Kriging. Zhu’s Concept of Geographic Similarity posits that “the more similar geographic configurations of two points (areas), the more similar the values (processes) of the target variable at these two points (areas).” This principle provides the theoretical foundation for spatial extrapolation, a critical aspect of spatial epidemiology that involves mapping disease risk by extending epidemiological patterns from isolated observations to entire regions.

Although geographic laws may not possess the precise mathematical definitions found in the physical sciences, they offer a valuable framework for understanding and analyzing geographic phenomena. These principles have been extensively validated and applied across various fields including natural ecology, human society, and economics [108,109,110,111]. Health, encompassing both human and animal health, is intrinsically a geographical phenomenon. Nonetheless, the application of geographic laws in health research, particularly epidemiology, has been approached with caution. This caution likely arises from the focus on living variables rather than non-living entities. In the medical field, establishing whether a geographic factor (e.g., particulate matter PM_2.5_) has a genuine impact on health necessitates experimental research, such as randomized controlled trials, clinical trials, or laboratory experiments involving cells or animals—these are considered the gold standards for determining causal relationships.

A significant risk of hypothesis-generating is falling into overly simplistic or misleading causal interpretations, a phenomenon known as the ecological fallacy [2,112]. Geographic correlation studies in epidemiology typically focus on large regions and broad geographic units—such as landforms, climate zones, land use, culture, economy, politics, and religion—alongside health group data. These studies aim to analyze population-level health risks and propose epidemiological hypotheses at the group level, rather than assessing individual health risks. The ultimate goal is to provide evidence-based recommendations and strategies for macro-level prevention policies, environmental management, planning, and healthcare resource allocation. However, individual health is influenced by a myriad of factors, including macro-geographic elements and individual-specific factors such as genetics, demographics (e.g., gender, age, race, income), and lifestyle. These individual-level factors may not be fully considered or are only indirectly represented in large- and medium-scale geographic analyses. Consequently, there is a risk of misinterpreting associations observed at the group level as applicable to individuals, which can lead to ecological fallacy. Some experts suggest that to make inferences from group to individual levels, geographic correlation studies should be conducted on local or small-scale levels to reduce ecological bias, as the analysis would be closer to the individual level [1]. Such analyses might ensure relative consistency in the geographic environment—potentially leveraging Tobler’s First Law of Geography—thereby allowing for a more nuanced examination of individual-level differences within a consistent environmental context. Alternatively, on a larger regional scale, selecting individuals with consistent individual-level characteristics (such as gender, age, and race) for observation can also be effective. In this scenario, geographic correlation studies may resemble natural cohort studies, case–control studies, or potential randomized controlled prevention or intervention trials, where the intervention arises naturally from different geographic or cultural environments rather than being artificially imposed.

To make geographic correlation analysis in spatial epidemiology more convincing, we propose the following two strategies:Control for individual-level variables while examining natural or cultural geographic factors affecting health spatial variation across large regions and scales. This approach aims to identify health differences and influencing factors among similar individuals in diverse geographic environments. This method requires the rigorous selection of individuals and study areas.Leverage the integrative nature of geography to conduct systematic studies on health influences, comprehensively observing factors across various scales/levels (e.g., global, continental, national, climate zones, watersheds, cities, communities, streets, individuals, etc.). This includes considering individual-level factors such as genetics and demographics, micro-scale factors like microclimates and community greening, meso-scale factors such as regional economy and transportation, and macro-scale factors like climate change, national trade, and ecosystems. This approach challenges the identification of sensitive levels and factors, data collection, and data processing capabilities.

The spatial epidemiological methods utilized in the context of swine viral diseases are broadly applicable across various types of epidemics and have seen widespread adoption [2,4]. The spatiotemporal pattern recognition techniques referenced in this paper demonstrate generalizability, with no apparent limitations when applied to other disease categories. However, given the unique transmission dynamics and pathways of each disease, the application of tools for risk factor identification necessitates careful consideration of disease-specific transmission characteristics. This includes the selection of appropriate influencing factors and the identification of key parameters within the mathematical models tailored to the particular epidemiological context.

## 7. Conclusions and Future Perspectives

The primary objectives of the application of spatial epidemiology in SVD encompass displaying and elucidating spatiotemporal distribution patterns, identifying spatial risk factors, risk mapping, and tracing the spatiotemporal evolution of pathogens. In this paper, we categorized various spatial analysis methods based on these objectives and provided detailed interpretations of their application in SVD. Special attention was given to considerations for reliable application and critical issues such as hypothesis generation. This comprehensive approach aims to enhance the broader and more sophisticated application of spatial epidemiology in the study and management of SVD.

With advancements in technologies such as artificial intelligence and evolving concepts like One Health, we outline the future directions for spatial epidemiology in the study of SVD:Syndemics research

Epidemics often result from the interplay of various environmental factors, including natural and social influences. Conversely, the integrated environmental context can influence the coexistence and spread of multiple epidemics, such as the co-infections of ASFV and CSFV. The study of synergistic effects among multiple epidemics has garnered significant attention, with spatial epidemiology offering a valuable technical framework for exploring the complex interactions of these epidemics within different natural and social contexts.

2.Temporal–spatial tracing research

Traditional epidemiological studies such as cross-sectional studies, case–control studies, and cohort studies rarely incorporate geographical location information into their study designs. They often rely on measuring the influence of backgrounds based on residential environments and static spatiotemporal factors [4].

In the field of non-communicable diseases, spatial lifecourse epidemiology [113] is gaining increasing attention. This approach aims to leverage advanced spatial and location-based technologies, such as earth observation, sensors, smartphone apps, and the Internet of Things (IoT), to investigate how environmental and other spatial factors (e.g., spatial accessibility) influence individual behaviors and health outcomes. We propose extending this concept to the spatial epidemiology of SVD by conducting continuous location-based monitoring of domestic pigs, farm pigs, and wild boars from the preconception period to disease endpoints. This would include tracking the transport and trade of pigs, as well as interactions between local pigs and wildlife. The focus shifts from merely responding to outbreaks to understanding prior exposures, which aids in identifying causes and determining the optimal times and locations for interventions.

This approach aligns closely with the One Health [114] philosophy, which recognizes the interconnectedness and interdependence of the health of humans, domestic and wild animals, plants, and the broader environment, including ecosystems. One Health emphasizes cross-sectoral, interdisciplinary, and cross-community collaboration at the human–animal–environment interface. When applying the One Health framework to SVD, research should focus on understanding the anthropogenic and ecological drivers of virus spillover from wild hosts to domestic pigs, identifying potential sources of contamination such as feed, water, vehicles, and personnel, and developing upstream, environment-targeted intervention strategies based on these insights. These efforts are inherently spatial and require the integration of spatial epidemiology techniques.

## Figures and Tables

**Figure 1 animals-14-02814-f001:**
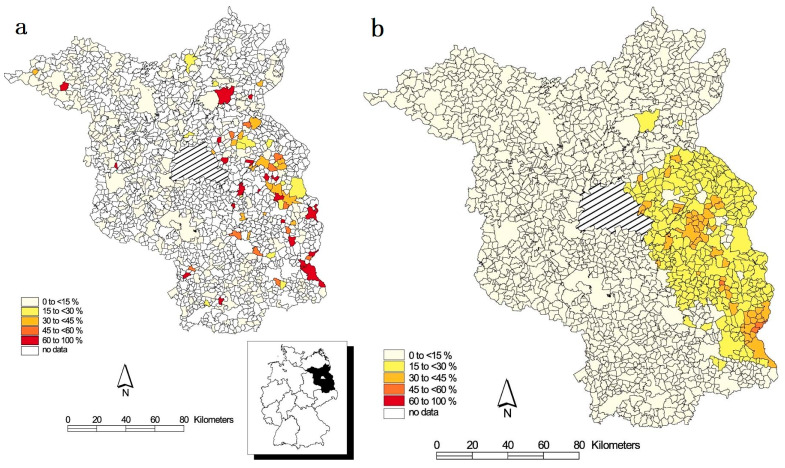
Observed prevalences of the PRV infections of wild boars in the Federal State of Brandenburg. (**a**) Estimated median prevalences for PRV infections of wild boars after smoothing using a BH model. (**b**) Figure reproduced from C. Staubach et al. (2002), with permission from Elsevier and the Copyright Clearance Center.

**Figure 2 animals-14-02814-f002:**
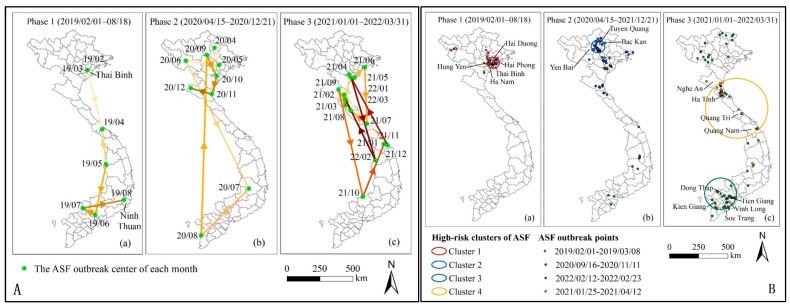
The monthly outbreak centers and average spread direction (**A**) and spatiotemporal high-risk clusters (**B**) of the ASF in Vietnam (February 2019–March 2022). Figure reproduced from Shao Qihui et al. (2022), licensed under an open access Creative Commons CC BY 4.0 license.

**Table 1 animals-14-02814-t001:** Commonly used spatiotemporal statistical methods and implementation software.

Spatial Data Type/Technology	Field Data Type	Advantage and Use	Software	Use in SVD
Point				
Mean Center	Discrete	Identify the geographic center (or the center of concentration) for a set of features	Spatial Statistics Tools (Esri)	A.S. Oganesyan et al. [20]
An algorithm proposed by the authors	Discrete	Discriminate between endemic and non-endemic patterns of case occurrence	Self-programming	M. Lange et al. [21]
Ripley’s K (Multi-Distance Spatial Cluster Analysis)	Continuous, discrete	Determine whether cases exhibit statistically significant clustering or dispersion over a range of distances	Spatial Statistics Tools (Esri); a macro in Excel	I. Iglesias et al. [22]; I. Iglesias et al. [23]; Satoshi Ito et al. [24]
Spatial or spatiotemporal clustering analysis	Continuous, discrete	Detect spatial or space–time disease clusters	SaTScan™, Matlab 2021a	I. Iglesias et al. [22]; Anastasia A. Glazunova et al. [25]; I. Iglesias et al. [23]; Satoshi Ito et al. [24]; Steven J. P. Tousignant et al. [26]; SeEun Choe et al. [27]; Pengfei Zhao et al. [28]
Local Colocation Quotient	Continuous, discrete	Measure local patterns of spatial association (collocated or not) between two categories of point features using the colocation quotient statistic.	Spatial Analyst Tools (ERSI)	Anastasia A. Glazunova et al. [25]
Spatial Autocorrelation (Moran’s I)		Measure spatial autocorrelation	Spatial Statistics Tools (Esri)	I. Iglesias et al. [23]
Direction analysis	Continuous, discrete	Determine whether the object tends to be in a systematic, directional spread	ClusterSeer software (Biomedware)	Shao, Qihui et al. [29]
Kernel density (Area also used)		Calculate a magnitude-per-unit area from point or polyline features using a kernel function to fit a smoothly tapered surface to each point or polyline	ArcGIS (ESRI, Redlands, USA); R (programming language)	Osvaldo Fonseca et al. [30]; Satoshi Ito et al. [24]; Beatriz Martínez-López et al. [31]; Nicolai Denzin et al. [32]; O. Berke et al. [33]; Nicolai Denzin et al. [34]
Directional Distribution (Standard Deviational Ellipse)		Provide the orientation and shape of a distribution, as well as its location, and dispersion or concentration of the data	ArcGIS (ESRI, Redlands, USA)	Osvaldo Fonseca et al. [30]; Satoshi Ito et al. [24]
Line				
Least Cost Path	Continuous, discrete	Find the shortest path between starting points and ending points across a cost surface	Spatial Analyst Tools (ERSI)	Fekede, R. J. et al. [35]
Network analysis		Shortest path length, clustering coefficient et al., applied to the analysis of livestock transport	R V.3.6.3, gephi v0.9.2	Alfredo Acosta et al. [36]
Area				
Hot Spot Analysis (Getis-Ord Gi*)		Identify statistically significant hot spots and cold spots	Spatial Statistics Tools (Esri)	Xin Pei et al. [37]
Spatial Autocorrelation (Moran’s I)		Measure spatial autocorrelation	Spatial Statistics Tools (Esri)	Xin Pei et al. [37]; Xiao Lu et al. [38]
Cluster and Outlier Analysis (Anselin Local Moran’s I)		Identify statistically significant hot spots, cold spots, and spatial outliers	Spatial Statistics Tools (Esri)	Xin Pei et al. [37]; Xiao Lu et al. [38]
Spatial or spatiotemporal clustering analysis	Continuous, discrete	Detect spatial or space–time disease clusters	SaTScan™, Matlab	Xin Pei et al. [37]; Pengfei Zhao et al. [39]; Xiao Lu et al. [38]; Nicolai Denzin et al. [32]; A. Allepuz et al. [40]; Lambert, M. E. et al. [41]; Thanapongtharm, W. et al. [42]; Huong, V.T.L. et al. [43]

**Table 2 animals-14-02814-t002:** Commonly used geographic correlation study methods and some indicative examples.

Methods		Use	Research Subjects	Geographic Environmental Factors Included	Identified Risk Factors	Use in SVD	Software
Statistical model	Cross-correlation analysis	Risk factors identification	CSF outbreaks	Climate variables (rainfall, wind speed, temperature, vapor pressure, and relative humidity).	Low relative humidity and high wind speed	Xiao Lu et al. [38]	SPSS
	Multivariable logistic regression		PRRSV serological status of pig farms	Farm size, geographic location, before or after the ASF outbreaks, and PRV purification	Pig farms	Pengfei Zhao et al. [39]	R
	Logistic mixed effect regression models		PRV exposure	Land cover	Agriculture; open-canopy pine, prairie, and scrub habitats	Hernández, F.A. et al. [46]	R
	Multivariable exact logistic regression model		PRRSV occurrence	Swine sites and trucking companies	Area spread (within three km) and truck network	Arruda, A.G. et al. [47]	SAS
	Generalized mixed-effects model		ASF case probability	Wild boar density, forest cover, built-up area, road density, population density, and proximity to previous infections	Wild boar density, near previous ASF incidents and forested areas	Podgorski, T. et al. [48]	R
	Bayesian hierarchical models		ASF occurrence	Wild boar density, densityof pig farms, road length (a proxy for human activity), and habitat availability	Wild boar density, road length (a proxy for human activity), and habitat availability	Depner, K. et al. [49]	R
	A mixed-effects Poisson regression model		ASF incidence rates	The number of pig farms, the number of pigs per commune, and human population density by commune	Pig density, pig farm density	Hien, N.D. et al. [50]	R
	Generalized Linear Logistic Regression		ASF outbreaks	Human and pig population, wild boarpopulation, settlements and smallholder farm distribution, legal movements of pigs and pork products, road networks, and forest areas	The importation of live pigs from ASF-affected regions, the density of smallholder farms, the volume of pork products’ importation from affected regions, the overall pig population, and the presence of a common border with an ASF-affected region	Glazunova, A.A. et al. [25]	R
	A generalized linear mixed model		Wild boar emergence	Animal emergence, altitude, slope, road density, human density, distance from water sources, wild boar distribution index, wild boar density, and capture pressure index	The appearance of raccoons, raccoon dogs, and crows as well as road density and wild boar distribution index	Ito, S. et al. [51]	R
	Bayesian spatial mixed multivariable logistic regression model		Probability of CSF occurrence	The number of farms, the number of pigs, pig movement, human demography, socioeconomic factors (household consumption, level of poverty), environmental factors (water sources, altitude, roads, and land cover)	Small family farms, high numbers of outgoing pig shipments and low levels of personal consumption	Martínez-López, B. et al. [52]	WinBUGS
	Zero-inflated generalized linear mixed-effects models		CSF infection risk	Municipality-level factors (season, habitat suitability, and the proportion of infected adjacent municipalities) and individual-level factors (age, sex, the proportion of infected cases, whether the wild boar was assigned to was vaccinated, and the cumulative number of time steps it was vaccinated)	The proportion of infected adjacent municipalities, age, month, and the proportion of infected cases	Scherer, C et al. [53]	R (package “glmmTMB”)
	Bayesian spatiotemporal hierarchical mode		PRRSV outbreaks	Environmental variables (weekly enhanced vegetation index, yearly averages of aboveground biomass density, canopy height, and land surface elevation; land surface temperature, and relative humidity), between-farm movement data (network metrics) and on-farm biosecurity features (site entries, perimeter buffer area access points, lines of separation access points, pig capacity, and farm density)	The animal movement network metric, out-degree, the number of lines of separation access points, number of days the temperature was between T [4 °C,10 °C], and relative humidity <20%	Sanchez, F. et al. [54]	R
Mathematical methods	Spatially explicit, individual-based (SEIB) model	Transmission process simulation and intervention effectiveness evaluation (scenario simulations)	The persistence of ASFV	Social structure of wild boar populations, dispersal between grid cells (a landscape unit)		Pepin, K.M. et al. [55]	Matlab
	Be-FAST (Monte Carlo approach)		The dynamic spread of CSFV	Proximity to infected pig farms, farmworker mobility, vehicular movements, pig and pig product transportation, climate variables, and weather patterns		Ivorra, B. et al. [56]	Matlab
	North American Animal Disease Spread Model (NAADSM)		The between-farm spread of PRRSV	Animal movement, truck sharing		Thakur, K.K. et al. [57]	NAADSM
	The SEIR model in a gridded landscape		The spatial dynamics of CSF outbreaks	The attraction of pig herds to water sources		Milne, G. et al. [58]	
	InterSprea (Monte Carlo approach)		The risk of introduction and spread of PRRSV	Pork importation, consumption patterns of pigmeat in households		Stevenson, M.A. et al. [59]	InterSpread Plus
Machine learning	The Extreme Gradient Boosting (XGBoost) machine learning model	Risk factor identification and risk prediction	The probability of PEDV infection	Animal movements, current PEDV status of farms, environmental factors (such as average temperature and humidity), and land use characteristics (including hog density and land use proportions)		Paploski, I.A.D. et al. [60]	Python
	Boosted regression trees (BRT)		PRRSV outbreaks	Pig farm, human population density and the number of farms with breeding sows		Thanapongtharm, W. et al. [42]	R

## Data Availability

The data presented in this study are available on request from the corresponding author(s).

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
