# Peer review of "Spatial Epidemiology and Its Role in Prevention and Control of Swine Viral Disease"

_animals, 2024, doi:10.3390/ani14192814_

Round 1

Reviewer 1 Report

Comments and Suggestions for Authors

Dear authors,

This is a very interesting, informative and comprehensive review, presenting and discussing the majority of techniques, methodologies and objectives for utilization and implementation of spatio-temporal epidemiological analyses. Most importantly, a context of the respective, arising needs that each of these methods and models serve is provided.

However, since most of them should summarize or be concluded by the provision, usually of a visual / graphical depiction of disease measures or status, perhaps some indicative figures, maps or illustrations could have been incorporated, just as an example of the available perspectives of spatial distribution of diseases. As you have already stated “the visual form of representation of health or disease distribution is more intuitive compared to numbers, tables, and text”!

A distinct reference and description of modelling tools is obtainable, nevertheless without further providing any insight into the model building process or performance of theirs. The reader in each case should resort to the initial referenced study or work for more detail. Perhaps, this would be beyond the scope of your study.

Additionally, it would be quite useful, at least to mention any potential extensions or implications (if any) of the application or utilization of such tools in other kind of diseases (i.e. bacterial, protozoal) and/or not only restrictive to swine species. Are there any crucial differences or similarities that should be considered? Vector borne diseases, that have beer used as paradigms in your study do not include only viral pathogens.

Bellow you will find some remarks and comments of the under-evaluation manuscript.

Lines 65-67 (a few typos): correct “discusse” to “discuss” and “offers” to “offer”.

Figure 1: Could you please explain your rationale for selecting the period interval of publications’ inclusion from 1992 to nowadays? Were there any specific criteria or assumptions for the opted starting point?

Table 1: There is a typo in the first area data type part; correct “identifie” to “identify”.

Line 185: What do you mean by “administrative units”? Could you please clarify?

Line 508: Use either “we advocate” or “we are advocating”, not “we advocating”.

Line 499-502: What are the individuals that you are referring to? What is the migration that they undergo? Are you referring to individual domestic or farmed pigs, herds or wild boars? Please clarify and elaborate about the life course approach of that statement.

Lines 503-506: I believe that the need, importance and contribution of spatial epidemiologic research, especially in viral diseases should extend beyond the timing and pathways of pig transportation / trade. The property of swine species of serving as an actual or potential reservoir and / or recombination vessel for many zoonotic viral diseases, along with the proximity and interaction with wildlife in many regions should not be overlooked, since it can be considered to constitute one of the main and constant public health concerns and threats.

As a general remark, I would suggest a more substantiated and unconstrained transition of your work and its multidisciplinary extensions to broader concepts of One Health and Planetary Health. It seems that the reference and participation in these integrated efforts to cope with contemporary challenges is performed in the margin of your study / manuscript. Indeed, spatial epidemiology, as presented in your study, offers a crucial and indispensable tool, yet I would recommend to elaborate in a more logical and natural way towards its association with the above concepts.

Comments on the Quality of English Language

These comments have been incorporated in the review report.

Author Response

Comments 1: However, since most of them should summarize or be concluded by the provision, usually of a visual / graphical depiction of disease measures or status, perhaps some indicative figures, maps or illustrations could have been incorporated, just as an example of the available perspectives of spatial distribution of diseases. As you have already stated “the visual form of representation of health or disease distribution is more intuitive compared to numbers, tables, and text”

Response 1: Thank you for pointing this out. We agree with this comment. Therefore, we have added indicative figures in 3.2.1.1. Disease Mapping (page 4) and 3.2.1.2. Spatiotemporal pattern recognition (page 8).

Comments 2: A distinct reference and description of modelling tools is obtainable, nevertheless without further providing any insight into the model building process or performance of theirs. The reader in each case should resort to the initial referenced study or work for more detail. Perhaps, this would be beyond the scope of your study.

Response 2: Agree. We have revised the manuscript accordingly, particularly Section 3.2.2, which previously provided a simple listing of models. We have removed the extensive descriptive passages and instead categorized and summarized various methods, their applications, and typical examples, presenting them in a tabular format to enhance clarity. This restructuring aims to provide readers with a comprehensive understanding of geographically related analytical methods, serving as a foundational reference that facilitates the selection of appropriate models for further in-depth study and research once their specific research objectives are defined.

Comments 3: Additionally, it would be quite useful, at least to mention any potential extensions or implications (if any) of the application or utilization of such tools in other kind of diseases (i.e. bacterial, protozoal) and/or not only restrictive to swine species. Are there any crucial differences or similarities that should be considered? Vector borne diseases, that have been used as paradigms in your study do not include only viral pathogens.

Response 3: Indeed, all the tools (methods) mentioned in the article are actually general and not limited to the field of swine viral diseases. They have been widely applied to other types of diseases, such as vector-borne diseases (e.g., malaria, schistosomiasis) and zoonotic diseases (e.g., brucellosis, rabies). Since this manuscript is a submission to a special issue focused on swine viral diseases, we did not extend the scope to other types of diseases in the main content. However, we have added our understanding of the broader applications of these methods in the discussion (Page 21).

Comments 4: Lines 65-67 (a few typos): correct “discusse” to “discuss” and “offers” to “offer”.

Table 1: There is a typo in the first area data type part; correct “identifie” to “identify”. Line 508: Use either “we advocate” or “we are advocating”, not “we advocating”.

Response 4: The spelling and grammatical errors have been corrected.

Comments 5: Figure 1: Could you please explain your rationale for selecting the period interval of publications’ inclusion from 1992 to nowadays? Were there any specific criteria or assumptions for the opted starting point?

Response 5: Thanks! We did not set a specific starting year for the keyword search. The year 1992 mentioned in the article refers to the earliest publication that could be retrieved using the keywords related to the topic of this study.

Comments 6: Line 185: What do you mean by “administrative units”? Could you please clarify?

Response 6: Yes, of course. Since epidemiological data is often collected and reported at administrative unit (e.g., community, town, city, state), spatial analysis is typically presented in the form of 'areas' corresponding to these units, such as prefecture-level or city-level divisions.

Comments 7: Line 499-502: What are the individuals that you are referring to? What is the migration that they undergo? Are you referring to individual domestic or farmed pigs, herds or wild boars? Please clarify and elaborate about the life course approach of that statement.

Response 7: In spatial lifecourse epidemiology, the term 'individual' refers to humans in the context of non-communicable diseases (NCDs). However, from a One Health systems perspective, it should encompass all entities related to a specific disease, including both humans and animals. In the case of swine viral diseases, this includes domestic pigs, farm pig herds, wild boars, and potentially all of them together. We have rewritten this section to make it clearer and more logically aligned with these concepts (Page 22).

Comments 8: Lines 503-506: I believe that the need, importance and contribution of spatial epidemiologic research, especially in viral diseases should extend beyond the timing and pathways of pig transportation / trade. The property of swine species of serving as an actual or potential reservoir and / or recombination vessel for many zoonotic viral diseases, along with the proximity and interaction with wildlife in many regions should not be overlooked, since it can be considered to constitute one of the main and constant public health concerns and threats.

Response 8: Indeed, it goes beyond just the timing and routes of pig transport and trade, which is only one aspect. We have added other factors, including the proximity to and interactions with wildlife, as you mentioned, among others (Page 22).

Comments 9: As a general remark, I would suggest a more substantiated and unconstrained transition of your work and its multidisciplinary extensions to broader concepts of One Health and Planetary Health. It seems that the reference and participation in these integrated efforts to cope with contemporary challenges is performed in the margin of your study / manuscript. Indeed, spatial epidemiology, as presented in your study, offers a crucial and indispensable tool, yet I would recommend to elaborate in a more logical and natural way towards its association with the above concepts.

Response 9: We agree with your suggestion. The outlook in the conclusion was indeed somewhat too brief and lacked fluency, which made it seem, as you mentioned, to have an unnatural transition. We have rewritten this section to improve its logical flow (Page 22).

Reviewer 2 Report

Comments and Suggestions for Authors

I have read and reviewed the manuscript entitled: “Spatial epidemiology and its role in prevention and control of Swine Viral Disease” with interest. It is an interesting topic. However, there are issues that need to be addressed before it is considered for publication.

 General comments

Review type: There are different types of reviews. However, it is not clear what type of review this manuscript is. The manuscript has a methods and results section, which may look like the systematic review style. However, systematic reviews have their own guidelines, including the Prisma Guideline (https://systematicreviewsjournal.biomedcentral.com/articles/10.1186/s13643-021-01626-4), which the current manuscript lacks. Authors should select one type of review and then rewrite the manuscript based on the specific review type that the authors choose to follow.

Here are some references:

1.        Narrative reviews: Green, B. N., Johnson, C. D., & Adams, A. (2006). Writing narrative literature reviews for peer-reviewed journals: Secrets of the trade. Journal of Chiropractic Medicine, 5(3), 101-117. doi:10.1016/S0899-3467(07)60142-6

2.        Systematic review and meta analysis: Moher, D., Liberati, A., Tetzlaff, J., Altman, D. G., & The PRISMA Group. (2009). Preferred Reporting Items for Systematic Reviews and Meta-Analyses: The PRISMA Statement. PLOS Medicine, 6(7), e1000097. doi:10.1371/journal.pmed.1000097

3.        Scoping reviews: Arksey, H., & O’Malley, L. (2005). Scoping studies: Towards a methodological framework. International Journal of Social Research Methodology, 8(1), 19-32. doi:10.1080/1364557032000119616

4.        Rapid review: Tricco, A. C., Antony, J., Zarin, W., et al. (2015). A scoping review of rapid review methods. BMC Medicine, 13, 224. doi:10.1186/s12916-015-0465-6

5.        Umbrella review: Fusar-Poli, P., & Radua, J. (2018). Ten simple rules for conducting umbrella reviews. Evidence-Based Mental Health, 21(3), 95-100. doi:10.1136/ebmental-2018-300014

6.        Grant, M. J., & Booth, A. (2009). A typology of reviews: An analysis of 14 review types and associated methodologies. Health Information & Libraries Journal, 26(2), 91-108. doi:10.1111/j.1471-1842.2009.00848.x

7.        Pawson, R., Greenhalgh, T., Harvey, G., & Walshe, K. (2005). Realist review—a new method of systematic review designed for complex policy interventions. Journal of Health Services Research & Policy, 10(1_suppl), 21-34. doi:10.1258/1355819054308530

Comments on the Quality of English Language

comments on language use aren't necessary at this stage of the manuscript.

Author Response

Comments 1: There are different types of reviews. However, it is not clear what type of review this manuscript is. The manuscript has a methods and results section, which may look like the systematic review style. However, systematic reviews have their own guidelines, including the Prisma Guideline (https://systematicreviewsjournal.biomedcentral.com/articles/10.1186/s13643-021-01626-4), which the current manuscript lacks. Authors should select one type of review and then rewrite the manuscript based on the specific review type that the authors choose to follow.

Here are some references:

  1. Narrative reviews: Green, B. N., Johnson, C. D., & Adams, A. (2006). Writing narrative literature reviews for peer-reviewed journals: Secrets of the trade. Journal of Chiropractic Medicine, 5(3), 101-117. doi:10.1016/S0899-3467(07)60142-6
  2. Systematic review and meta analysis: Moher, D., Liberati, A., Tetzlaff, J., Altman, D. G., & The PRISMA Group. (2009). Preferred Reporting Items for Systematic Reviews and Meta-Analyses: The PRISMA Statement. PLOS Medicine, 6(7), e1000097. doi:10.1371/journal.pmed.1000097
  3. Scoping reviews: Arksey, H., & O’Malley, L. (2005). Scoping studies: Towards a methodological framework. International Journal of Social Research Methodology, 8(1), 19-32. doi:10.1080/1364557032000119616
  4. Rapid review: Tricco, A. C., Antony, J., Zarin, W., et al. (2015). A scoping review of rapid review methods. BMC Medicine, 13, 224. doi:10.1186/s12916-015-0465-6
  5. Umbrella review: Fusar-Poli, P., & Radua, J. (2018). Ten simple rules for conducting umbrella reviews. Evidence-Based Mental Health, 21(3), 95-100. doi:10.1136/ebmental-2018-300014
  6. Grant, M. J., & Booth, A. (2009). A typology of reviews: An analysis of 14 review types and associated methodologies. Health Information & Libraries Journal, 26(2), 91-108. doi:10.1111/j.1471-1842.2009.00848.x
  7. Pawson, R., Greenhalgh, T., Harvey, G., & Walshe, K. (2005). Realist review—a new method of systematic review designed for complex policy interventions. Journal of Health Services Research & Policy, 10(1_suppl), 21-34. doi:10.1258/1355819054308530

Response 1: Thank you for your valuable suggestions. We have carefully reviewed the references you provided, examined various exemplary reviews, and considered the objectives of our article. Based on this, we have identified our review as a narrative review. The reasons are as follows:

  1. Our aim is to provide a comprehensive overview of the current applications of spatial epidemiology in swine viral diseases. This review is intended to offer a broad perspective for veterinarians and public health decision-makers without a background in geography, enabling them to quickly understand how spatial analysis can contribute to the control of swine viral diseases. Our focus is on summarizing the types of applications, tools used in the field, and the potential biases in current applications, providing important considerations when using these tools. Finally, we discuss future trends in this field, particularly in relation to the "One Health" approach, highlighting the need for broader interdisciplinary collaboration that extends beyond geography and epidemiology.
  2. Spatial epidemiology is an interdisciplinary field between geography and epidemiology. In such an interdisciplinary field, where theories and methods from one discipline (geography) are applied to address practical issues in another discipline (epidemiology), we believe that a rigorous risk of bias assessment and evidence quality evaluation may not be necessary. Instead, a narrative review would be more appropriate.

   We have revised the structure and content of the paper according to the format of a Narrative Review.